# Multi-Dimensional Impact of Chronic Low Back Pain among Underserved African American and Latino Older Adults

**DOI:** 10.3390/ijerph18147246

**Published:** 2021-07-06

**Authors:** Mohsen Bazargan, Margarita Loeza, Tavonia Ekwegh, Edward K. Adinkrah, Lucy W. Kibe, Sharon Cobb, Shervin Assari, Shahrzad Bazargan-Hejazi

**Affiliations:** 1Department of Family Medicine, Charles R Drew University of Medicine and Science (CDU), Los Angeles, CA 90059, USA; mohsenbazargan@cdrewu.edu (M.B.); MLoeza@mednet.ucla.edu (M.L.); shervinassari@cdrewu.edu (S.A.); 2Department of Public Health, CDU, Los Angeles, CA 90059, USA; edwardadinkrah1@cdrewu.edu; 3Physician Assistant Program, CDU, Los Angeles, CA 90059, USA; lucykibe@cdrewu.edu; 4Department of Family Medicine, UCLA, Los Angeles, CA 90095, USA; 5School of Nursing, CDU, Los Angeles, CA 90059, USA; tavoniaekwegh@cdrewu.edu (T.E.); sharoncobb@cdrewu.edu (S.C.); 6Department of Psychiatry, UCLA, Los Angeles, CA 90095, USA; 7Department of Psychiatry, CDU, Los Angeles, CA 90059, USA

**Keywords:** back pain, chronic, underserved, African American, Latino, older adults

## Abstract

Chronic low back pain is one of the most common, poorly understood, and potentially disabling chronic pain conditions from which older adults suffer. The existing low back pain research has relied almost exclusively on White/Caucasian participant samples. This study examines the correlates of chronic low back pain among a sample of underserved urban African American and Latino older adults. Controlling for age, gender, race/ethnicity, education, living arrangement, and number of major chronic conditions, associations between low back pain and the following outcome variables are examined: (1) healthcare utilization, (2) health-related quality of life (HR-QoL) and self-rated quality of health; and (3) physical and mental health outcomes. Methods: We recruited nine hundred and five (905) African American and Latino older adults from the South Los Angeles community using convenience and snowball sampling. In addition to standard items that measure demographic variables, our survey included validated instruments to document HR-QoL health status, the Short-Form McGill Pain Questionnaire-2, Geriatric Depression Scale, sleep disorder, and healthcare access. Data analysis includes bivariate and 17 independent multivariate models. Results: Almost 55% and 48% of the Latino and African American older adults who participated in our study reported chronic low back pain. Our data revealed that having low back pain was associated with three categories of outcomes including: (1) a higher level of healthcare utilization measured by (i) physician visits, (ii) emergency department visits, (iii) number of Rx used, (iv) a higher level of medication complexity, (v) a lower level of adherence to medication regimens, and (vi) a lower level of satisfaction with medical care; (2) a lower level of HR-QoL and self-assessment of health measured by (i) physical health QoL, (ii) mental health QoL, and (iii) a lower level of self-rated health; and (3) worse physical and mental health outcomes measured by (i) a higher number of depressive symptoms, (ii) a higher level of pain, (iii) falls, (iv) sleep disorders, (v) and being overweight/obese. Discussion: Low back pain remains a public health concern and significantly impacts the quality of life, health care utilization, and health outcomes of underserved minority older adults. Multi-faceted and culturally sensitive interventional studies are needed to ensure the timely diagnosis and treatment of low back pain among underserved minority older adults. Many barriers and challenges that affect underserved African American and Latino older adults with low back pain simply cannot be addressed in over-crowded EDs. Our study contributes to and raises the awareness of healthcare providers and health policymakers on the necessity for prevention, early diagnosis, proper medical management, and rehabilitation policies to minimize the burdens associated with chronic low back pain among underserved older African American and Latino patients in an under-resourced community such as South Los Angeles.

## 1. Background

A recent study examining the most common global conditions found that low back pain ranked as the fourth most common condition in primary care visits in developed countries [1]. Indeed, chronic low back pain is one of the most common, poorly understood, and potentially disabling chronic pain conditions from which older adults suffer [2]. A review of the literature clearly shows that both the incidence and prevalence of severe and chronic low back pain increase with older age [3]. There is evidence that older adults with chronic low back pain may be at a higher risk of cardiovascular disease [4], serious falls [5], low HR-QoL [6,7,8], depression [9], poorer postural control and interference with activity [10,11], low physical activity levels [12], functional and mobility impairments [13], diminished energy capacity and slower gait speed [14], kinesiophobia [15], a low sense of self-efficacy [16], disability [17], appetite impairment [18], substance use [19], and sleep disorders [20]. Older adults living with low back pain often describe their golden years as “not so golden after all” [21].

The health disparity among African Americans and Latinos has been recognized and documented in the literature. People of color often fare worse than their White counterparts across many measures of health status such as life expectancy and the burden of chronic disease. However, existing low back pain research has relied almost exclusively on White/Caucasian participant samples [22,23,24,25]. Knowledge on the impact of chronic low back pain among underserved African American and Latino older adults is limited. Examination of baseline data from the Back pain Outcomes using Longitudinal Data (BOLD) registry (the largest inception cohort to date of seniors presenting to a primary care provider for back pain) shows that being of the African American race and older age were associated with worse functional disability as measured by the Roland–Morris Disability Questionnaire [22]. Additional empirical data suggest that there are racial disparities in the use of opioids for back pain among older adults [26]. Furthermore, one nationally representative sample of Medicare claims showed that the provision of care for uncomplicated low back pain varied according to nonclinical factors: mainly demographic characteristics, socioeconomic status, and area of residence. These findings highlight the need to ensure that the provision of care is based on clinically relevant characteristics and is not biased by sociodemographic factors [27].

This study examines the correlates of the chronic low back pain among a sample of underserved urban African American and Latino older adults. Controlling for age, gender, race/ethnicity, education, living arrangement, and number of major chronic conditions, associations between low back pain and the following outcome variables are examined: (1) physical HR-QoL; (2) Mental HR-QoL; (3) depression symptoms; (4) insomnia/sleep difficulty; (5) self-rated health status; (6) number of Rx medication used; (7) adherence to drug regimens; (8) complexity of medication; (9) falls; (10) ED utilization; (11) hospital admissions; (12) office-based provider visits; and (13) level of pain.

Since previous studies have clearly demonstrated that an array of independent and interdependent relationships exist between social determinants of health and chronic low back pain [28] and given our limited understanding of chronic low back pain among underserved minority older adults, there is a clear need to systematically explore the prevalence, correlates and conditions that may contribute to worse outcomes among this segment of our population. Given the nature of diseases in minority groups, findings from our study will add to the evidence used in making public health decisions. Information on the risk factors of chronic low back pain is essential for policymakers. Understanding the impact of chronic low back pain is crucial for healthcare resource planning among these populations.

## 2. Methods

### 2.1. Recruitment

We recruited nine hundred and five (905) participants from senior centers, senior housing centers, faith-based organizations, and apartment complexes in the South Los Angeles community using convenience and snowball sampling. We visited 52 sites throughout the study. We included participants who self-identified as African American or Latino aged 55 years or older and who committed to being in the area for at least one year. Recruitment was completed during regular office hours, usually in the day room or dining room areas. Face-to-face structured interviews were terminated in early February 2020 due to the COVID-19 pandemic. Home to over one million residents, South Service Planning Area 6 (SPA6) of Los Angeles County is disproportionately harmed by health disparities compared to the rest of Los Angeles County [29].The study was approved by the Charles R. Drew University of Medicine and Science (CDU) Institutional Review Board.

### 2.2. Measurements

Survey instruments: In addition to the standard items that measure demographic variables, our survey included validated instruments to document HR-QoL [30], health status, the Short-Form McGill Pain Questionnaire-2 (SF-MPQ-2) [31,32], depression (GDS) [33,34], sleep disorders, [35,36] and healthcare access.

Demographics characteristics: We used age, gender, educational attainment, marital status, and race/ethnicity as the covariates in this study. Educational attainment was obtained as a continuous variable (number of years). Higher scores indicated more years of education. We asked our participants whether they were married or lived with a partner, which was analyzed categorically as either married/lived with a partner or not married/do not live with a partner. We also asked our participants whether they lived alone or if there was any other member of the family such as a partner or a spouse who lived with them, which was analyzed categorically as either living alone or living with at least 1 other individual.

Financial strain: This variable was measured using five items: “In the past 12 months, how frequently were you unable to: (1) buy the amount of food your family should have, (2) buy the clothes you feel your family should have, (3) pay your rent or mortgage, (4) pay your monthly bills (5) make ends meet?” Items were on a 5-level response scale ranging from 1 (never) to 5 (always). A total “financial strain” score was calculated, with an average score of five items, ranging from 1 to 5. A high score was indicative of greater financial difficulty. These items are consistent with Pearlin’s list of chronic financial difficulties experienced by low SES individuals (Cronbach alpha = 0.92) [37].

Accessibility of medical care: This variable was measured using five items: (1) In the last 12 months, have you needed to see a doctor but could not see one? (2) Overall, how difficult is it for you to get medical care? (3) How difficult would it be for you to get a routine physical exam? (4) How difficult is it for you to visit a doctor when you need medical care? (5) Overall, how satisfied are you with how available medical care is for you? Principal component analysis was used to identify potential factors underlying this 5-item instrument. Only one factor was produced, which was able to explain almost 50% of the variance. All items had primary loadings over 0.4. A lower score on this index reflects a higher level of perceived difficulty accessing medical care.

ED Utilization, Office-Based Provider Visits, and Hospital Admissions: Participants were asked how many times they had utilized ED, stayed overnight at hospital as a patient, and visited a provider at their office in the last 12 months.

Fall incidents: Fall incidents were assessed with one single item asking participants to report a single fall or multiple falls in the year before the interview (yes = 0; no = 1).

Major chronic conditions: Participants reported whether they have been diagnosed with any of the following conditions: high blood pressure, diabetes mellitus, heart-related conditions, cancer, stroke, and/or chronic obstructive pulmonary disease (COPD).

Self-rated Health Status (SRH): Participants reported SRH using a single question: “In general, would you say your health is (1) Excellent; (2) Very good; (3) Good; (4) Fair (5) Poor?” Knowing that self-rated health is differently shaped by social determinants across ethnic groups, it is important to include perceived-self status as one of the indices that measures the health condition of participants [38]. In addition, this item is repeatedly used in large-scale national surveys and predicts mortality risk among Latino and other ethnic groups [39,40]. However, National Health Interview Surveys indicate that self-rated health predicts mortality risk less well for older Hispanics than their non-Hispanic White counterparts [41].

Pain Severity: This study employed the definition as stated by the American Academy of Family Physicians, which stipulates low back pain (LBP) as, “pain, muscle tension, or stiffness localized below the costal margin and above the inferior gluteal folds, with or without sciatica, and is defined as chronic when it persists for 12 weeks or more” [42]. Low back pain was measured using the four subscales outlined in the Short-Form McGill Pain Questionnaire-2 (SF-MPQ-2) [31,32]. Participants self-reported the level to which they experienced each of the four subscales, which are composed of 22 pain items experienced in the past week using an 11-point numeric rating scale. Examination of the psychometric properties of the SF-MPQ-2 among Hispanic and non-Hispanic White patients with pain shows that this measure seems to be used equivalently across these 2 ethnic groups [43]. This pain subscale used in our sample had high internal reliability (Cronbach’s alpha = 0.950 in the African American population; Cronbach’s alpha = 0.961 in the Latino population; Cronbach’s Alpha = 0.952 in both the Latino and African American populations) and excellent validity (*p* < 0.05, 95% CI [−2.39294, −1.85743]).

Health-Related Quality of Life Survey (SF-12): The SF-12 Health Survey measures include two subscales that measure mental and physical functioning and overall health-related quality of life [30]. The SF-12 is a multipurpose short-form survey with 12 questions, all selected from the longer SF-36 Health Survey [44]. The SF-12 is a validated measure for assessing the health status of low-income minority populations [45]. We computed the physical and mental health composite scores (PCS & MCS) using twelve questions that have scores ranging from 0 to 100, where a 0 score indicates the lowest level of health and 100 indicates the highest level of health. The PCS and MCS scores tend to vary by age: PCS tends to decrease while MCS tends to increase with age. The age-specific mean difference score (difference score) is the amount by which a person’s score differs from their age group’s mean score. For individual scores, someone who scores higher than the mean indicates a person with better health status than most others their age. Conversely, scores lower than the mean indicate a person with poorer health than most others their age [46].

Depressive Symptoms: This study used the 15-item Short Geriatric Depression Scale (GDS) to evaluate depression [33,34]. Responses were assessed by “yes” or “no” responses. A summary score was calculated with a potential range between 0 and 15, in which a higher score indicated more depressive symptoms. The GDS-Short Form has excellent reliability and validity, and it has been used extensively to measure depression among older adults in both community and clinical settings [33,34].

Sleep Quality: Sleep quality was measured using the 7-item Insomnia Severity Index (ISI) [35,36]. Each item was on a Likert scale with possible answers ranging from 0 to 4. We calculated a total score ranging from 0 to 28, with a higher score reflecting worse sleep quality. Research shows that the ISI has adequate validity and internal consistency (reliability) to evaluate sleep difficulties.

Medication Use: Medication use was assessed by recording an inventory of all the medications each participant was taking within two weeks prior to the interviews. Participants were asked to bring all over-the-counter and prescribed medications to the interview. From the container label, he interviewer transcribed the name of the medication, strength of the drug, expiration date, instructions, special warnings, providers’ information, etc. This medication methodology, established by Sorensen and colleagues [47,48,49] has been adopted by our research team previously [24,50,51,52,53,54].

Medication regimen complexity index (MRCI): This study employed the MRCI, a tool for quantifying multiple features of drug regimen complexity. The tool was developed and validated by George and colleagues [55]. Medication regimen complexity is a theoretical concept independent of pharmacologic, clinical, and demographic factors. The MRCI quantifies the complexity of medication regimens according to dosage forms, dosing frequencies, and additional directions. The MRCI is an open index without an upper limit for the number of drugs that could be prescribed to a patient or the number of additional instructions possible in a particular regimen. It is a reliable and valid tool with potential applications in clinical practice and research [55]. All components of the MRCI have been independently confirmed to influence patient adherence [56].

Blood Pressure: An OMRON HEM 907XL Intellisense Professional blood pressure monitor was used to measure participants’ blood pressure. Our team physician and education specialist measured the blood pressure of all of the participants. Clinically validated for accuracy, the OMRON HEM 907XL Intellisense Professional blood pressure monitor has been designed for clinical use in a professional setting. All blood pressure measures were taken twice with one minute between measures. An average of the two measures was recorded. The monitor was calibrated twice during the study according to the manufacturer’s instructions.

Body-Mass Index (BMI): The survey included a self-report of height and weight. Participants were classified according to their BMI as: normal weight (BMI 20.0–24.9 kg/m^2^); overweight (BMI 25–29.9 kg/m^2^); or obese (BMI 6 30 kg/m^2^).

### 2.3. Data Analysis

Our analysis had three parts. The first section was a descriptive analysis. This descriptive work reported the means and standard deviation for the continuous measures and the frequency and percentages for the categorical variables. Second, we conducted Pearson correlation coefficients, an independent *t*-test, and ANOVA to examine the bivariate association between socio-demographic variables, other relevant variables, and low back-pain. Finally, we used (1) multivariate generalized linear models (GLM) with Poisson distribution and log link; (2) multiple linear regression; and (3) multiple binary and multinomial logistic regression to examine the independent association of low back pain on various outcomes. Controlling for age, gender, race/ethnicity, education, living arrangement, and number of major chronic conditions, the association between low back pain and several outcome variables was examined. For GLM and logistic regression, odds ratio (OR) and 95% confidence intervals are reported, and for liner regression, standardized beta coefficients are reported. For multivariate analysis, *p* values of less than 0.05 were considered significant. The Bonferroni correction method was used to counteract the problem of multiple comparisons between the bivariate association.

## 3. Results

Table 1 (1st column) reports the characteristics of the study sample. This study included 740 (81.8%) African American and 165 (18.2%) Latino individuals who were between the ages of 55 and 96 years of age (mean = 71.50 ± 8.36). Approximately 34% of the participants were 75 years of age or older, with 44% self-reporting living alone. One-third of the sample never completed high school. Regarding health status, we noted the following health conditions/illnesses: diabetes mellitus (38%), hypertension (89%), heart-related conditions (28%), and COPD (27%). The majority of the participants (79%) were overweight or obese, with 44% having a BMI of 30 and over.

### 3.1. Bivariate Analysis:

Table 1 (column 2–4) shows bivariate correlations between low back pain and other related variables. These correlations are based on the chi-square, *t*-test, and ANOVA F-test. Female, younger, less educated, and obese participants reported more low back pain than their counterparts. In unadjusted models, associations between reporting low back pain and the following variables were detected: falls, sleep disorders, dissatisfaction with healthcare, a higher number of office-based physician visits, ED and hospital admissions, a lower level of HR-QoL, a higher number of depressive symptoms, a lower level of adherence with medication, a higher number of Rx medication use, and a lower level of self-reported health status. No associations between, low back pain and systolic blood pressure, over the counter (OTC) medication use, living arrangement, and ethnicity were detected.

### 3.2. Multivariate Analysis

Table 2 contains the condensed results of 17 independent multivariate analyses. Each analysis controlled for age, gender, race/ethnicity, living arrangement, financial strain, and number of major chronic conditions while examining the independent impact of low back pain on 17 outcome variables. This table reveals that having low back pain was associated with three categories of outcomes: (1) a higher level of healthcare use measured by (i) physician visits, (ii) ED admissions, (iii) a higher level of Rx use, (iv) a higher level of medication complexity, (v) a lower level of adherence to medication regimens, and (vi) a lower level of satisfaction with medical care; (2) a lower level of QoL and self-assessment of health measured by (i) a lower level of physical health QoL, (ii) a lower level of mental health QoL, and (iii) a lower level of self-rated health; and (3) worse physical and mental health outcomes measured by (i) a higher number of depressive symptoms, (ii) a higher level of pain, (iii) falls, (iv) sleep disorders, (v) and being obese.

## 4. Discussion

Almost 55% and 48% of Latino and African American older adults in our study reported chronic low back pain, a much higher prevalence than the national average for White, Black, and Hispanic older adults. Data from the latest (2018) National Health Interview Survey (NHIS) revealed that the prevalence of low back-pain among White, Black, and Hispanic men aged 55 years and older during the preceding 12 months was 35.2%, 34.7%, and 32.1% respectively. The prevalence for White, Black, and Latino women in a similar age group was 37.3%, 40.0%, and 40.3%, respectively. Our findings show a much higher prevalence of low back pain for both African American and Latino underserved men and women compared to NHIS data for the same age group. In our study, the prevalence for African American and Latino men was 43.6% and 40.4% and for women, it was 50.2% and 61.1%, respectively. It is very important to note that our data was collected from one of the most underserved and under-resourced areas in the State of California known as South Service Planning Area 6 (SPA6) of the County of Los Angeles. The much higher prevalence of low back pain among our sample of Latinos and African Americans points to urgent interventions among this and other underserved communities of minority older adults.

Low Back Pain and Healthcare Utilization: Our findings indicate that African American and Latino older adults who suffer from low back pain used ED services more frequently and had more office-based provider visits than their counterparts with no low back pain. More than 47% of African Americans and 31% of Latinos aged 55 years and older who suffered from low back pain used ED services and were admitted to the hospital at least once within the 12 months prior to the interview. Multivariate analysis showed that after adjusting for age, gender, education, living arraignment, financial strain, and number of chronic conditions, participants with low back pain had 49% increased odds of using ED services, whereas admission to the hospital was no longer significant. Further, multivariate analysis revealed that after adjusting for other variables, participants with low back pain had 51% odds of visiting office-based providers at least every two months compared to their counterparts with no back pain.

A systematic review of the literature indicates that low back pain is consistently a top presenting complaint in the emergency setting [57]. The National Electronic Injury Surveillance System shows that among all individuals in the US presenting with low back pain in emergency departments, older patients were found to be at a greater risk of hospital admission for the treatment of low back pain [58]. However, several studies documented racial disparities in the treatment of low back pain in both the ED and in the primary care setting [26,59,60,61,62,63,64]. National data show that controlling for potential confounders, non-White patients who presented at the ED for back pain were less likely than their White counterparts to receive analgesia and more likely to wait longer for opiate medication [60]. One recent study conducted by Kohen and colleagues (2018) among 600 adult patients with low back pain from three ED departments in Michigan revealed that racial disparities and psychosocial factors had concerning relationships with clinical decision-making in the ED among patients with low back pain, indicating that Caucasian race was one factor associated with advanced imaging [59].

Low Back Pain and Medication Use and Adherence to Drug Regimens: Our study documented a strong association between low back pain and the use of Rx but not OTC. On average, African American and Latino older adults with low back pain used almost one (0.89) additional prescription medication than their counterparts without low back pain. In addition, we documented that, after adjusting for other relevant factors, non-adherence to drug regimens is more prevalent among our sample of underserved African American and Latino older adults with low back pain. Medication-related challenges, including polypharmacy (excessive and unnecessary use of medication), inappropriate medication use, and non-adherence to drug regimens among minority adults, particularly among African Americans, have recently received attention among researchers [53,65,66,67,68,69,70,71,72]. Limited available data showed increased medication-related challenges among minority older adults compared to their White counterparts [53,65,66,67,68,69,70,71,72]. Polypharmacy remains an important issue among underserved older minority adults [54]. African Americans with polypharmacy, particularly those with hyper-polypharmacy, are experiencing higher levels of psychological distress, which itself is a known risk factor for poor adherence to medications [73]. Polypharmacy also has been linked to depressive symptoms in U.S.-born older Mexican Americans [74].

Low Back Pain and Accessibility of Medical Care: Another interesting result of this study was that at both the bivariate and multivariate levels, participants with low back pain were more likely to express their dissatisfaction with the availability of and satisfaction with medical care than their counterparts with no back-pain. Minority older adults forgoing needed medical care due to barriers associated with cost, type or lack of health care coverage, mistrust, racism, and discrimination are at risk of missing the needed care that may be necessary for the management of pain and other chronic conditions [75]. The sources of disparities in pain management among ethnic and racial minorities are complex, involving patient, health care provider, and health care system factors [76]. Recent findings (2020) noted that pain is often poorly recognized, inadequately assessed, and unsuccessfully managed among people in mainstream society, but this is particularly the case for people who have been historically, economically, and socially marginalized [77]. A qualitative study by Wallace and colleagues (2021) suggested that pain is entangled with and shaped by experiences of inadequate and ineffective health care as well as experiences of discrimination, stigma and dismissal, and the impacts of these and other factors intersecting experiences [78]. Equity-oriented responses to chronic pain would recognize pain not only as a biomedical issue but as a social justice issue [78] and therefore improving access and satisfaction with medical care among minority older adults with pain is the first step toward mitigating some of the racial/ethnic disparities observed in pain among this segment of our population.

Low Back Pain and Falls: Multivariate analysis of our data revealed that, after adjusting for other variables, participants with low back pain had 62% odds of having experienced a fall at least once within last 12 months compared to their counterparts with no back pain. The Centers for Disease Control and Prevention (CDC) reports that falls are currently the leading cause of both fatal and non-fatal injury and trauma among older adults [79]. Previous studies show that older adults with a higher number of chronic diseases were more likely to have fallen in the past year compared to those with fewer chronic diseases [80]. In addition, multiple studies have documented that older adults with chronic low back pain are at risk for falling [81,82,83,84].

While extensive research has been conducted on falls and fall-related risk factors, the majority is primarily among non-Hispanic White populations. There are mixed findings in the literature in regard to racial differences and falls. Although some studies have found that older non-Hispanic Whites are more likely to fall than older African Americans [85,86,87], a few revealed no racial differences in fall rates among these two groups [88,89,90]. However, examining ethnic differences in fall rates and fall circumstances in older community-dwelling White and African-American women, Faulkner and colleagues documented that although the circumstances of falling differed for older White and African American women, there were no differences in the frequencies of falling [88]. Similarly, using California Health Interview Survey data, another recent study reported that African American older adults were most likely to experience ≥ 2 falls in the past year (14.2%), followed by Hispanics (13.8%), non-Hispanic Whites (12.8%), and Asian Americans (7.6%) [80]. This study argued that further research is needed to explore factors associated with fall risks across racial/ethnic groups in order to design culturally appropriate interventions to reduce falls among this segment of our population [80]. Knowing that African American and Latino older adults suffer from the poor management of multiple conditions [91] and have a greater number of chronic conditions, including untreated or under-treated low back pain, higher and more severe falls among this segment of our population are inevitable if no appropriate interventions are implemented.

Low Back Pain and Sleep Disorder: Our study showed that, after adjusting for other relevant variables, including the number of chronic conditions, the association between low back pain and sleep disorders remained significant. The association between low back pain and sleep complaints (such as disruptive sleep and problems initiating and maintaining sleep) among older adults has been well-stablished [2,92]. However, the examination of sleep disorders among minority older adults are very limited [93,94,95,96,97]. Furthermore, there are few studies evaluating sleep quality among minority older adults and its relationship with pain [96]. A recent study conducted among 1664 older adults revealed significant racial/ethnic differences in overall sleep quality and in individual sleep components among older adults [93]. Interventional studies show that different types of therapies are effective in reducing the pain related to disability, quality of sleep, and depression in older adults with chronic pain and low back pain [98,99]. However, future studies are needed to understand the role of different social, psychological, physical as well as pharmacological therapy in relation to sleep disturbance and low back pain among African American older adults. Sleep evaluation should be routinely included in studies of under-resourced minority older adults with chronic pain, particularly if they have low back pain.

Low Back Pain and Depression: Our data revealed a strong association between the Geriatric Depression Scale (GDS) and low back pain, even after adjusting for other relevant variables, among underserved African American and Latino older adults. A review of the literature shows that pain is a consistent risk factor for the development of depressive symptoms [100], and the presence of chronic pain is an independent contributor to depressive symptoms, even when controlling for each individual chronic illness, and the overall number of illnesses [101]. A review of epidemiologic studies conducted by Zis and colleagues showed that among older adults, chronic pain increases the risk of depression between 2.5 and 4.1 times. Moreover, patients suffering from a major depressive disorder are six times more likely to suffer from neuropathic pain and three times more likely to suffer from non-neuropathic pain [102]. Zis and colleagues hypothesized that the common pathogenic factor between chronic pain and depression could be represented by chronic, subclinical neuroinflammation [102]. The risk of disabling back pain increases in older age [9,103,104].

There are very limited studies that have examined the association between pain or low back pain and depression among minority older adults [105,106]. Data from the Health and Retirement Study (HRS) showed that severe pain was more strongly associated with a greater risk of depressive symptoms among Black and Hispanic older adults than among White older adults. The increased risk of mental disorders that racial and ethnic minorities face due to severe chronic pain is particularly important in the context the aging process [107]. Data from Community Aging in Place: Advancing Better Living for Elders (CAPABLE) showed that depression fully mediated the relationship between pain intensity and functional limitation among low-income older adults [105]. Another study showed that beyond the obvious physical manifestations and limitations of the pain experience, there was a robust relationship between depression and pain among a sample of African American women [106].

Low Back Pain and Blood Pressure: Our study documented no association between low back pain and blood pressure among our sample of underserved African American and Latino older adults. The association between hypertension and low back pain has not been clearly documented. There are a few of studies that have demonstrated an inverse relationship between high blood pressure and the prevalence of low back pain and osteoarthritis [108]. Indeed, it is still unclear how hypertension and hypertensive medicine influence sensitivity to pain. However, it is stablished that pain sensitivity is frequently diminished in hypertensive individuals. A European study showed that in a large unselected population sample with a very wide age range, the presence of chronic pain was associated with decreased heart rate variability and baroreflex sensitivity relative to the absence of chronic pain, supporting a model in which the risk of hypertension in the chronic pain population is derived, in part, from the diminished heart rate variability and baroreflex sensitivity associated with chronic pain [109]. More definitive studies on the relationship between hypertension, low back pain, and osteoarthritis, particularly among older minority adults, are needed.

Low Back Pain and Quality of Life: While several studies have documented an association between pain and quality of life [6,8,110,111,112,113,114,115], the current study used a community-based sample of under-resourced African American and Latino older adults to document the significant association between chronic low back pain and both the physical and mental health quality of life among this segment of our population. Cedraschi and colleagues (2016) investigated the self-reporting of low back pain and associated health-related quality of life in a population sample of 3042 community-dwelling older adults aged ≥65. They suggested that low back pain may be a risk factor for frailty, defined by higher levels of functional limitations, psychological difficulties, and social restrictions, hence, globally impaired HRQoL [116]. Several cross-sectional and longitudinal studies conducted in European countries have also provided very similar results, indicating a strong association between low back pain and HR-QoL [7,117,118]. Finally, a recent systematic reviews of 35 research paper published from 1985 to 2018 with a total sample of 135,059 older adults aged between 60 and 102 years confirmed a high prevalence of low back pain in older adults and functional disability that affects quality of life and factors important for independence [119]. It is important to note that this systematic review found a significant absence of studies among older Latino and African Americans. More studies on the effect of low back pain on quality of life are needed in this population.

Low Back Pain and Body Mass Index: Accumulative evidence from longitudinal and cross-sectional studies link the occurrence of musculoskeletal diseases, including chronic low back pain, to the presence of obesity and being overweight [120,121,122]. Our study shows that 66% of African American and Latino older participants with a BMI ≥ 30 reported low back pain versus 34% with a BMI < 25. The association between low back pain and BMI remains significant even after adjusting for other related variables. The epidemic of obesity and being overweight remains a heavy burden among minority populations. Crespo and colleagues argued that although obesity and its consequences affect all age groups and ethnic subgroups, minority older adults are disproportionately affected [123]. A cross-sectional analysis of adults over 65 in the 2013 National Health Interview Study (NHIS) (N = 7714) showed that Non-Hispanic Whites (25%) had the lowest percentage of obesity compared to Latinos (27%) and African Americans (36%) [124]. Specifically, high obesity rates persist across the lifespan of African American women, from young to older adulthood, with class 3 (severe) obesity present at nearly twice the rate in women and men of other races in older adulthood [125]. A third National Health and Nutrition Examination Survey that involved 5724 adults over 60 shows that the prevalence of significant back pain increases markedly with increased BMI, particularly among African American and Mexican American older adults [126]. The relationship between BMI and back pain was stronger among Mexican Americans (44% increase from normal weight 12% to obesity class III 31%) and African Americans (43% from normal weight 29% to obesity class III 23%) than Whites (12% increase from normal weight 21% to obesity class III 24%) [126]. Our study along with limited available data for older minority adults regarding the association between low back pain and BMI point to the urgent need for interventional studies to address the relationship between being overweight and pain (knee, back, and hip) among older minority adults.

## 5. Study Limitations

Several limitations were noted in this study. First, the cross-sectional research design as employed in this study attributes no causal relationships between chronic low back pain and the variables analyzed. While chronic low back pain bears a significant relationship with fall risk, a vice versa association may also exist where the risk of falls impacts chronic low back pain. Another limitation of this study is that we did not collect any data regarding occupational histories prior to the onset of low back pain. Participants who have experienced years of heavy labor may be more likely to suffer from low back pain, and this may influence the outcomes and conclusions of this research. Additionally, participants were required to recall their history of chronic low back pain without providing and/or reviewing their medical records. This could introduce recall bias. Additionally, data was obtained using a convenience sampling method; results may either under- or over-represent the Latino and African American sampling frame. Despite these limitations, the findings contribute to the body of knowledge on chronic low back pain in under-resourced Latino and African American populations.

## 6. Implications

From this study, underserved African American and Latino older adults suffering from chronic low back pain are more likely to use emergency department services, prescription medication(s), express dissatisfaction with medical care, and be obese. When further compared with their White counterparts, they are also likely to have depressive symptoms, poor physical and mental quality of life, and experience sleep disorders as well as to fall over the course of 12 months. However, an association with blood pressure readings within this population cannot established through this study.

Our study contributes to and raises awareness of clinicians, healthcare providers, and health policymakers on the necessity for prevention, early diagnosis, proper medical management, and rehabilitation policies to minimize the burdens associated with chronic low back pain among underserved older African American and Latino patients in under-resourced communities. To treat this complex musculoskeletal disorder, there needs to be a multidisciplinary approach of resource allocation and training with a focus on health equity, mental health, and healthcare access. Objectively measuring the quality of life of minority older adults suffering from chronic low back pain patients is necessary to establish objectives and treatment plans to prevent the progression of recurring and worsening chronic lower back pain. Primary care providers can make a difference in the overall outcomes and improvements of health equity and back health. This can be accomplished in a variety of ways through (1) providing recommendations for back health, (2) health promotion initiatives that focus on proper body mechanics, (3) expansion of chiropractic services, and (4) expansion of data analytics to translate data into actionable information to guide and inform standardized public policy on musculoskeletal health.

At the population level, it is important to monitor the evolution of chronic low back pain using data such as socio-demographic characteristics, health care utilization, health care satisfaction metrics, and physical and mental health outcomes to assess the impact on older Black and Latino adults. At the individual level, chronic low back pain is more than a musculoskeletal disease; it affects all aspects of the lives of minority adults, including mental health, quality of sleep, health-related quality of life and BMI as well as health care utilization and adherence to drug regimens. Primary care physicians are often the first to identify minority older patients with chronic low back pain and can have a significant impact on their care. A multi-disciplinary and inter-professional approach to chronic low back pain should be an integral part of care to prevent disability and improve quality of life among minority older adults. These interventions may mitigate inappropriate ED visits and reduce health care expenditures. These health care resources can then be redirected to the prevention of chronic low back pain and the improvement of the quality of life of older adults. More research is needed to study the impact of chronic low back pain on the quality of life, health care utilization, and health outcomes among minority older adults in underserved communities.

## Figures and Tables

**Table 1 ijerph-18-07246-t001:** Characteristic of sample a bivariate associations between chronic low back pain and other related outcomes (n = 905).

Sample Characteristics	Total Sample	Chronic Low-Back Pain		Sig.
		No	Yes	
	N (%) Mean ± SD	N (%) Mean ± SD	N (%) Mean ± SD	
**Gender**				0.009
Male	317 (35.0)	178 (57)	135 (43)	
Female	588 (65.0)	277 (48)	303 (52)	
**Race/Ethnicity**				0.113
Hispanics	165 (18.2)	70 (45)	85 (55)	
Non-Hispanic Blacks	740 (81.8)	385 (52)	353 (48)	
**Living Arrangement**				0.087
Living Alone	397 (44)	215 (54)	182 (46)	
Live with other(s)	496 (56)	240 (48)	256 (52)	
**Falls in last 12 months**				0.006
No	357 (71)	155 (44)	195 (56)	
Yes	145 (29)	45 (31)	100 (69)	
**Sleep Disorder**				0.000
No	646 (72)	382 (59)	264 (41)	
Yes	247 (28)	73 (30)	174 (70)	
**BMI**				0.023
<25.0	103 (20)	42 (42)	59 (58)	
25–29.9	178 (35)	83 (48)	91 (52)	
≥30	224 (44)	75 (34)	145 (66)	
**ER Admissions**				0.000
No	548 (61)	307 (57)	234 (43)	
Yes	354 (39)	148 (42)	204 (58)	
**Hospital Admissions**				0.002
No	663 (73)	354 (54)	301 (46)	
Yes	241 (27)	101 (42)	137 (58)	
	**Mean ± SD**	**Mean ± SD**	**Mean ± SD**	**Sig.**
**Age** (Years: 55–96)	71.5 ± 8.36	72.6 ± 7.88	70.2 ± 8.64	0.000
**Education** (Years: 1–16)	11.9 ± 3.32	12.12 ± 3.20	11.7 ± 3.35	0.046
**Financial Strains** (Alway:1–Rarely: 5)	4.05 ± 1.22	4.36 ± 1.02	3.74 ± 1.31	0.000
**Major Chronic Conditions** (0–6)	2.08 ± 1.11	1.91 ± 1.04	2.27 ± 1.14	0.000
**Systolic blood pressure**	133 ± 20.0	135 ± 20.5	132 ± 20.0	0.095
**Satisfaction with Medical Care**	2.72 ± 0.95	2.65 ± 1.00	2.84 ± 0.95	0.003
**Physical Health Quality of Life**	40.1 ± 12.2	45.1 ± 10.71	34.9 ± 11.40	0.000
**Mental Health Quality of Life**	51.9 ± 11.6	54.4 ± 9.41	49.4 ± 13.05	0.000
**Depressive Symptoms**	2.79 ± 2.90	1.98 ± 2.31	3.63 ± 3.20	0.000
**Severity of Pain**	2.06 ± 2.30	1.05 ± 1.52	3.17 ± 2.46	0.000
**Number of Rx Used**	5.75 ± 3.25	5.31 ± 3.16	6.20 ± 3.28	0.000
**Number of OTC Used**	1.11 ± 1.84	1.13 ± 1.77	1.08 ± 1.91	0.659
**Sleep Disorder index**	4.99 ± 6.09	3.50 ± 4.84	7.11 ± 7.02	0.000
**Self-rated Health**	3.17 ± 1.01	2.90 ± 1.01	3.45 ± 0.95	0.000
**Physician Visits**	5.32 ± 3.21	4.88 ± 3.18	5.80 ± 3.19	0.000
**ED Admissions**	0.75 ± 1.50	0.54 ± 1.13	0.97 ± 1.79	0.000
**Hospital Admissions**	0.71 ± 1.27	0.42 ± 1.23	0.61 ± 1.32	0.026
**Adherence with Medication**	5.88 ± 2.13	6.32 ± 1.85	5.61 ± 2.25	0.001
**Medication Complexity**	11.67 ± 7.23	10.55 ± 6.77	12.83 ± 7.49	0.000

**Table 2 ijerph-18-07246-t002:** Multivariate analysis of associations between chronic low back pain and outcome variables.

Dependent/Outcome Variable	Model	Independent Variable: Chronic Low Back Pain (No vs. Yes) Adjusting for Age, Gender, Education, Living Arraignment, Financial Strains and Number of Chronic Conditions					
		B	Beta	95% CI Unstandardized Coefficients	Exp. (B) OR	95% CI Exp. (B) OR	Sig.
**Health Care Utilization**							
Physician Visits	MLR	0.717	0.112	0.282–1.151			0.001
ED Admissions	GLM-P				1.490	1.152–1.928	0.002
Hospital Admissions	GLM-P				1.269	0.872–1.848	0.214
Number of Rx Used	MLR	0.508	0.078	0.103–0.914			0.014
Number of OTC Used	GLM-P				1.016	0.807–1.280	0.892
Medication Complexity	MLR	1.569	0.109	0.664–2.475	N/A	N/A	0.001
Adherence with Medication	MLR	−0.410	−0.094	−0.808–−0.012	N/A	N/A	0.044
Satisfaction with Medical Care	MLR	0.133	0.071	0.009–0.257			0.035
**QoL Self–rated Health**							
Physical Health Quality of Life	MLR	−7.593	−0.312	−9.033–−6.153	N/A	N/A	0.000
Mental Health Quality of Life	MLR	−3.102	−0.134	−4.616–−1.588	N/A	N/A	0.000
Self–rated Health	MLR	0.364	0.179	0.233–0.494	N/A	N/A	0.000
**Physical and Mental Health Outcomes**							
Systolic Blood Pressure	MLR	−2.325	−0.057	−6.168–1.517	N/A	N/A	0.235
Depressive Symptoms	MLR	0.881	0.152	0.538–1.224	N/A	N/A	0.000
Severity of Pain	LT-LR	0.563	0.409	0.485–0.640	N/A	N/A	0.000
Falls in last 12 months No Yes	BLR	N/A	N/A	N/A	1.000 1.619	Ref 1.056–2.483	0.004
**Sleep Disorder** No Yes	BLR	N/A	N/A	N/A	1.000 2.66	Ref 1.721–4.117	0.000
**BMI** <25 25–30 >30	MNLR	N/A	N/A	N/A	0.863 0.858 1.00	0.518–1.438 0.407–0.968 Ref	0.573 0.035

MLR: Multiple linear regression; LT-LR: log transferred linear regression; BLR: binary logistic regression; MNLR: multinomial logistic regression; GLM-P: generalized linear regression with Poisson distribution.

## Data Availability

Personal identification details of the participants were separated from the completed questionnaires. The data were stored in a locked room of the corresponding author at the Charles R. Drew University of Medicine and Science (CDU). No information relating to identifiable individuals was disseminated. The data sets used and analyzed in the current study are available from the corresponding author for collaborative studies.

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
