# Peer review of "Multi-Dimensional Impact of Chronic Low Back Pain among Underserved African American and Latino Older Adults"

_ijerph, 2021, doi:10.3390/ijerph18147246_

Round 1
Reviewer 1 Report
The topic of pain in the lumbar spine is a well-known and researched topic. As the authors note, the topic has been studied mainly in the Caucasian race. The analysis of other groups (African American and Latino older adults) is a good complement. This is a novel topic that will be of interest to the readers of the journal. The work was written very cleverly and legibly. The reader is gradually introduced to the research topic. The discussion was interesting and comprehensive. I suggest some improvements for this study.
In the introduction, it would be worth mentioning the state of knowledge about inequalities in healthcare between ethnic groups. Later references to it are frequent.
Line 107-109 - such information should not be included in the description of the recruitment of participants. It should be included in the discussion
It is necessary to shorten the discussion. data in the discussion that were already presented in the results should not be repeated. The discussion should be redrafted.
In the implications section, take what is the most important from this research and then describe the general recommendations. There is no synthetic summary of what was obtained with this study.
The limitations of the study are not listed.
Author Response
AUTHORS’ RESPONSE TO REVIEWER 1
Comment 1: The topic of pain in the lumbar spine is a well-known and researched topic. As the authors note, the topic has been studied mainly in the Caucasian race. The analysis of other groups (African American and Latino older adults) is a good complement. This is a novel topic that will be of interest to the readers of the journal. The work was written very cleverly and legibly. The reader is gradually introduced to the research topic. The discussion was interesting and comprehensive. I suggest some improvements for this study.
Response 1: Thank you for your time and thoughtful feedback
Comment 2: In the introduction, it would be worth mentioning the state of knowledge about inequalities in healthcare between ethnic groups. Later references to it are frequent.
Response 2: The second paragraph of the manuscript’s introduction/background describes these inequalities.
Comment 3: Line 107-109 - such information should not be included in the description of the recruitment of participants. It should be included in the discussion.
Response 3: Lines 107 to 109 have been removed and considered in the discussion section.
Comment 4: It is necessary to shorten the discussion. Data in the discussion that were already presented in the results should not be repeated.
Response 4: The discussion has been shortened and redrafted. Repetitive material already presented in the study’s results section have also been removed from the discussion.
Comment 5: In the implications section, take what is the most important from this research and then describe the general recommendations. There is no synthetic summary of what was obtained with this study.
Response 5: The first paragraph of the implication section synthetically summarizes the most important findings from this study. General set of recommendations are further presented in continuing paragraphs of the implications section
Comment 6: The limitations of the study are not listed.
Response 6: Thank you for bringing this to our notice. The limitation section has been clearly included in the manuscript
Warm regards,
Shahrzad Bazargan-Hejazi

Reviewer 2 Report
The title “Multi-dimensional impact of chronic low-back pain among underserved african American and latino older adults”conducted by Bazargan and colleagues recruited and investgate 905 African American and Latin senior adults,collected a lot of their personal information and emphatically compared the relationship between chronic pain and some related factors,such as survey instruments, denographics characteristics, financial strain, medical care, pain severity, depressive symptoms and so on. The factors considered are detailed and comprehensive.
This manuscript has a strong logic, and his highlight is that a lot of detailed discussions on various comparative factors in the discussion section.
It would be a good comparison factor to know whether these old people have a history of high-intensity labor work.
Author Response
AUTHORS’ RESPONSE TO REVIEWER 2
Comment 1: The title “Multi-dimensional impact of chronic low-back pain among underserved African American and Latino older adults “conducted by Bazargan and colleagues recruited and investigate 905 African American and Latin senior adults collected a lot of their personal information and emphatically compared the relationship between chronic pain and some related factors such as survey instruments, demographics characteristics, financial strain, medical care, pain severity, depressive symptoms and so on. The factors considered are detailed and comprehensive.
Response 1: Thank you for your comments and feedback
Comment 2: This manuscript has a strong logic, and his highlight is that a lot of detailed discussions on various comparative factors in the discussion section
Response 2: Thank you very much
Comment 3: It would be a good comparison factor to know whether these old people have a history of high-intensity labor work.
Response 3: We did not collect any data regarding these two important “potential” predictors of low-back pain among older adults. This has been acknowledged in the limitations section of the manuscript.
Warm regards,
Shahrzad Bazargan-Hejazi

Reviewer 3 Report
- Well-written article is an important contribution to our understanding of LBP with its focus on African American and Latinx older adults, underserved populations often not included in LBP research. There is, however, a relative lack of detail about workplace/occupational history related to LBP, which makes the inclusion of this paper in a special issue of LBP and Occupational Safety and Health. Do the authors have occupational-related data to include in this article? For example, can the authors include employment status as a variable? This information may be especially of importance as inclusion criteria recruited persons as young as age 55.
- Abstract - Conclusion mentions African American older adults, but not Latinx - include both give the focus on the analysis?
- Keywords - Lation spelling.
- Lines 74-76 - While Medicare research demonstrates less utilization of diagnostic imaging in older, ethnic/racial minority, and low income patients with LBP, this may not (necessarily) be problematic, as clinical practice guidelines discourage the use of diagnostic imaging in most LBP cases - the issue guideline-congruence might be noted by the author.
- Instruments - Include psychometric properties for African Americans as well as Latinx populations; include information about psychometrics in older adult populations as able - self-rated health and pain measures in particular.
- Sleep measures - Consider editing this section for length - all items need not be specified as currently listed.
- LBP - How was low back pain defined and measured? This key variable was not described in the methods section...the definition of the LBP variable is of critical importance to this study. What distinction was made between acute or chronic LBP?
- Line 246 - with typo.
- Table 1/2 results suggest the participants used many medical resources in the past year, but in the introduction and discussion, the participants were described as 'underserved' - can this discrepancy be resolved? In what way are these participants underserved?
- Line 282 - Discussion notes 'a much lower prevalence' of LBP in study participants, but the statistics then provided for other epidemiological studies suggest a higher prevalence. Please clarify.
- Lines 310-313 - Authors cite paper that shows racial/ethnic minority patients receive less prescription medicine and surgery, and use more complementary treatments for pain. However, this is actually LBP guideline-congruent care (see ACP guidelines for LBP, which recommend non-pharmacological treatments as first line treatment over medicines and surgery). This perhaps inadvertent guideline-congruence must be discussed.
- Depression - Very extensive discussion. Is this level of detail needed?
- References - Non-standard referencing formats with some references repeated in bibliography.
Author Response
AUTHROS RESPONSE TO REVIEWER 3
Comment 1: Well-written article is an important contribution to our understanding of LBP with its focus on African American and Latinx older adults, underserved populations often not included in LBP research. There is, however, a relative lack of detail about workplace/occupational history related to LBP, which makes the inclusion of this paper in a special issue of LBP and Occupational Safety and Health. Do the authors have occupational-related data to include in this article? For example, can the authors include employment status as a variable? This information may be especially of importance as inclusion criteria recruited persons as young as age 55.
Response 1: Unfortunately, we did not collect any data regarding these two important “potential” predictors of low-back pain among older adults. This has been acknowledged in the limitations section of the manuscript
Comment 2: Conclusion mentions African American older adults, but not Latinx - include both give the focus on the analysis?
Response 2: This was a typographical omission of the Latino population. This has been rectified in the manuscript’s
Comment 3: Keywords - Lation spelling.
Response 3: This typographical error has also been corrected; Lation - Latino
Comment 4: Lines 74-76 - While Medicare research demonstrates less utilization of diagnostic imaging in older, ethnic/racial minority, and low income patients with LBP, this may not (necessarily) be problematic, as clinical practice guidelines discourage the use of diagnostic imaging in most LBP cases - the issue guideline-congruence might be noted by the author.
Response 4: The congruence is duly noted and based on the citation referenced, this section has been redrafted to highlight the existence of biases noted within Medicare claims in the provision of care. These biases vary based on non-clinical factors as described in the manuscript.
Comment 5: Instruments - Include psychometric properties for African Americans as well as Latinx populations; include information about psychometrics in older adult populations as able - self-rated health and pain measures in particular.
Response 5: Psychometric properties of pain measure as identified in both African American and Latino populations have been computed and included in the methods section of the manuscript. This pain subscale utilized in our sample had high internal reliability (Cronbach's alpha = 0.950 in African American population; Cronbach's alpha = 0.961 in Latino population) and validity.
Comment 6: Sleep measures - Consider editing this section for length - all items need not be specified as currently listed.
Response 6: Section on measures of sleep has been reviewed and items removed from the manuscript.
Comment 7: LBP - How was low back pain defined and measured? This key variable was not described in the methods section...the definition of the LBP variable is of critical importance to this study. What distinction was made between acute or chronic LBP?
Response 7: The definition of low back pain according to the American Academy of Family Physicians, is pain, muscle tension, or stiffness localized below the costal margin and above the inferior gluteal folds, with or without sciatica. They also defined low back pain as chronic when it persists for 12 weeks or more. This in addition to the low back pain measure has been included in the manuscript.
Comment 8: Line 246 - with typo”.
Response 8: Typographical error corrected
Comment 9: Table 1/2 results suggest the participants used many medical resources in the past year, but in the introduction and discussion, the participants were described as 'underserved' - can this discrepancy be resolved? In what way are these participants underserved?
Response 9: The qualitative and quantitative lack or scarcity of resources exists within service planning area (SPA) 6 where our sampling took place relative to other SPAs within LA County. This difference as further captured in the findings of this study lists various constructs that are disproportionately lacking.
Comment 10: Line 282 - Discussion notes 'a much lower prevalence' of LBP in study participants, but the statistics then provided for other epidemiological studies suggest a higher prevalence. Please clarify.
Response 10: The statement should rather indicate a higher prevalence and not lower. This has been duly rectified.
Comment 11: Lines 310-313 - Authors cite paper that shows racial/ethnic minority patients receive less prescription medicine and surgery, and use more complementary treatments for pain. However, this is actually LBP guideline-congruent care (see ACP guidelines for LBP, which recommend non-pharmacological treatments as first line treatment over medicines and surgery). This perhaps inadvertent guideline-congruence must be discussed.
Response 11: This section has been duly corrected and redrafted. The congruence noticed is likely to due to the lack of emphasis on chronic low back pain treatment in our initial submission. The guidelines for the treatment of acute vs chronic back pain differs.
In 2017 the American College of Physicians (ACP) recommended in an evidence-based clinical practice guideline published in Annals of Internal Medicine that physicians and patients should treat acute or subacute low back pain with non-drug therapies such as superficial heat, massage, acupuncture, or spinal manipulation. If drug therapy is desired, physicians and patients should select nonsteroidal anti-inflammatory drugs (NSAIDs) or skeletal muscle relaxants. (https://www.acponline.org/acp-newsroom/american-college-of-physicians-issues-guideline-for-treating-nonradicular-low-back-pain)
Comment 12: Depression - Very extensive discussion. Is this level of detail needed?
Response 12: The discussion on depression has been shortened to a more a reasonable level of applicable detail.
Comment 13: References - Non-standard referencing formats with some references repeated in bibliography.
Response 13: References have been standardized using MDPI styling format and duplicates removed.
Warm regards,
Shahrzad Bazargan-Hejazi

Round 2
Reviewer 1 Report
Thank you for resending the article for review. I analyzed the corrections made by the authors. The article may be accepted for publication in this form. I have no more comments.